# Radiofrequency Lesion in the Atrial Wall: How Variable Is It? 9.4 Tesla MRI Analysis of Radiofrequency Lesion Volume in a Swine Model

**DOI:** 10.3390/jcm13175153

**Published:** 2024-08-30

**Authors:** Laura Sofia Cardelli, Thomas Laumont, July Beghian, Yosra Achahli, Maida Cardoso, Marylène Bacle, Jean-Luc Pasquié, Mathieu Granier

**Affiliations:** 1Département de Cardiologie, Centre Hospitalier Régional Universitaire de Montpellier, 34295 Montpellier, France; 2Cardiology Department, Versilia Hospital, 55041 Camaiore, Italy; 3Phymedexp, Université de Montpellier, Inserm, CNRS, CHRU de Montpellier, 34295 Montpellier, France; 4BNIF Facility, Montpellier University, 34093 Montpellier, France; 5Faculté de Médecine, Université de Montpellier, RAM-PTNIM, 30900 Nîmes, France

**Keywords:** radiofrequency ablation, radiofrequency energy, cardiac arrhythmias, magnetic resonance imaging, 9.4 Tesla

## Abstract

**Background/Objectives**: Most data on radiofrequency (RF) effects come from ex vivo or in vitro studies that quantify lesions using width and/or depth, while electrophysiologists use manufacturers’ indirect indices. The objective of this study was to evaluate RF lesion volume by high-resolution MRI of excised lesions in an in vivo porcine model, comparing a low-energy long-duration (LE) (20 W, 50 s) RF application strategy with a high-energy short-duration (HE) (50 W, 20 s) one. **Methods**: Eighteen piglets were divided into LE (n = 9) and HE groups (n = 9). RF applications were performed at four locations in both atria. Animals were sacrificed after 5–7 days, and RF lesion specimens were excised, fixed, and analyzed by 9.4 Tesla MRI. RF lesion volume, variability (variance), depth, and any extracardiac lesions were compared between the groups. **Results**: Seventy RF applications were performed (36 LE, 34 HE). MRI analyzed 26 LE and 28 HE samples. The HE group showed 35% higher volume than the LE group (100.2 mm^3^ (±81.2) in LE vs. 178.3 mm^3^ (±163.7) in HE, *p* = 0.033). RF volume variance was 6.6 mm^3^ in LE and 40.3 mm^3^ in HE. The HE group had more complications (seven vs. zero, *p* = 0.02) and extracardiac lesions (18 vs. 14, *p* = 0.613). **Conclusions**: There was large and unpredictable variability in RF injury on the atrial wall, even under controlled conditions, which could explain arrhythmia recurrences. The greatest lesion variability was found during HE applications. The dose/effect relationship of RF needs careful study for treating cardiac arrhythmias.

## 1. Introduction

Radiofrequency (RF) energy has been widely used for the ablative treatment of cardiac arrhythmias since the early 1990s [1]. Effective RF lesions require continuous, transmural, or deep destruction of myocardial tissue both on the atrial and ventricular sides [2,3,4]. However, several questions remain about the optimal dose of RF energy needed to ensure definitive injury, without delivering excessive energy, which could increase the risk of extracardiac injury.

Ablation catheter manufacturers have proposed indices such as the Ablation Index and the Lesion Size Index (LSI) [5,6] to help physicians deliver sufficient energy to achieve transmural lesions. Most of these indices have been validated in ex vivo studies, either on the ventricular wall or via an epicardial approach in animal models [6,7,8]. Additionally, several studies have attempted to quantify RF lesion size using indirect indices, such as width and depth, or by comparing lesion geometry with various power settings [9,10]. As a result, the available studies are highly variable and often incomparable. The exact volume of RF lesions remains largely unknown, particularly the variability in RF lesions under clinical conditions.

New imaging methods, including photoacoustic and magnetic resonance imaging (MRI), have provided further insight into the characterization of myocardial tissue [11] and RF-induced morphological changes [12,13,14,15,16,17,18]. Unlike the histopathologic method, which allows for only unidirectional cutting of the specimen and is limited by the number of planes measured, MRI allows for multiplane evaluation and three-dimensional (3D) reconstruction, making the evaluation of the entire lesion volume more accurate. 

Even though RF has been studied as part of ablative strategies for atrial fibrillation, especially in recent combinations of high- and low-energy with long vs. short delivery duration, there is a lack of direct data regarding the exact lesion volume produced. Data are often derived from in vitro animal studies [19], or even in silico models [9]. Human data mostly concern clinical safety endpoints and atrial arrhythmia recurrences [20,21,22]. 

In this study, we aimed to compare, using 9.4 Tesla MRI, the volume of RF lesions produced at the atrial level and the extra-cardiac effects in two groups of swine, one treated with a low-energy/long-application (20 Watts, 50 s) strategy and another treated with high-energy/short-application (50 Watts, 20 s) strategy.

## 2. Materials and Methods

### 2.1. Study Protocol

The experimental protocol was approved by the regional animal research ethics committee (approval: APAFIS#23000-2019112614376164v31736). It was conducted on 18 large white piglets (weighing 28 ± 2.3 kg) anesthetized by intramuscular injection of ketamine (10 mg/kg) and atropine (0.05 mg/kg), intubated, and mechanically ventilated. After induction, sedation was maintained with propofol infusion (15 mg/kg/h) and additional bolus of sufentanyl (0.1 µg/kg) when required.

Echo-guided punctures of the right femoral were performed to introduce an 8.5-F long sheath (SL0, Abbott medical) to perform transseptal puncture (BRK XS, Abbott medical transseptal needle). Through a 6-F sheath, a quadripolar deflecting diagnostic catheter (Xtrem, Sorin, Milano, Italy) was introduced into the coronary sinus, and a 6-lead surface (4 limbs, 2 precordial) electrocardiogram was used to monitor heart rhythm. All electrical signals were processed with a bandpass filter from 30 to 500 Hz and digitally recorded (CardioLab II plus 32, GE Healthcare, Chicago, IL, USA). Unfractionated heparin (100 IU/kg) was administered after vascular punctures.

### 2.2. Ablation Procedure

Animals were divided into 2 groups as follows: (1) a low-energy (LE)/long-application group, with RF power at 20 W for a 50 s duration (n = 9), and (2) a high-energy (HE)/short-application group, with RF power at 50 W for a 20 s duration (n = 9). 

An 8 Fr contact force irrigated catheter (Tacticath Abbott medical, St Paul, MN, USA), with saline irrigation of 17 mL/min during the application, was used for RF deliveries using a dedicated generator (IBI 1500-T11, Abbott medical, St Paul, MN, USA) connected to the digital recording system. This allowed for continuous monitoring of the following RF application parameters: power, impedance, temperature, duration, Force–Time Integral (FTI), and LSI. 

In each case, 4 widely separated RF application spots were performed, by a single operator, to allow for easy individualized excision and measurement of lesion volume by MRI. Two RF applications were performed on the posterior wall of the right atrium (RA), near the superior vena cava (RAs) and inferior vena cava (RAi). After RF application in the right atrial wall, transseptal puncture was performed, and the following applications were made in the left atrium (LA): one on the superior (LAs) and one on the inferior (LAi) part (Figure 1). The objective was to achieve a minimum contact force of 10 gr and a maximum of 20 gr. RF was applied on plane surfaces as much as possible.

Electrogram (EGM) signal amplitude and bipolar and unipolar signals were recorded immediately before and after RF application (Figure 2). 

Complications of the procedure (such as steam pop during RF application, tamponade that required pericardial drainage, ventricular arrhythmia, or death) were recorded.

### 2.3. Sacrifice and Lesion Volume Measurement

After the ablation procedure, the swines were extubated and followed for 5 to 7 days to allow for recovery of RF injury edema. After this time, the animals were euthanized by injection of pentobarbital (70 mg/kg bolus). After sternotomy, the heart and lungs were excised and flushed. The aorta, pulmonary artery, esophagus, and right and left lungs (the upper and middle lobes) were carefully inspected to detect any RF lesion by contiguity. The RA and LA were then incised as follows: along the anterior wall between the superior and inferior vena cava for RA and between the two pulmonary veins for LA. Each RF lesion area was individually excised, rinsed, and macroscopically analyzed (recording specimen thickness, flat/trabecular surface) (Figure 3) before staining in 4% paraformaldehyde (PFA). The specimens were then transported to MRI for measurement of lesion volume.

### 2.4. MRI Processing

Given the small size of tissue samples, 1.5 to 3 Tesla MRIs were unable to achieve adequate histological accuracy. Therefore, we decided to perform MRI acquisitions with a 9.4 Tesla MRI at the Laboratoire Charles Coulon (Institute of Physics, CNRS and University of Montpellier). The MRI acquisitions were performed on post-mortem specimens after the heart samples were placed in a 10 mm diameter glass tube filled with Fluorinert FC-770 liquid (3M™ Electronic Liquids, Saint Paul, MA, USA) to keep specimens in good hydration conditions without interfering with MRI. 

Subsequently, the glass tube was placed in a custom-made coil in the 9.4 Tesla MRI (Agilent Varian 9.4/160/ASR, Santa Clara, CA, USA). Given the heterogeneity in the samples, 3D acquisition was the best method for measuring RF lesion volumes. After testing different MRI sequences, T2-weighted 3D Spin Echo was the most appropriate to determine RF lesion volume (Figure 4). 

After the acquisition, 3D MRI visualizations and segmentations were performed using Myrian Software (Intrasense, Montpellier, France). This allowed for the acquisition of RF lesion volume (mm^3^) and depth (mm) data.

### 2.5. Statistics

Normally distributed values were presented as mean and standard deviation (SD) and compared by T-tests and one-way analysis of variance; otherwise, the median value (interquartile range [IQR]), the Mann–Whitney U test, and the Kruskal–Wallis test were used. Categorical variables were summarized in terms of counts and percentages and compared using the two-sided Pearson’s chi-squared test. To reduce the influence of outliers, lesion volume data were subjected to winsorization at the 1st and 99th percentiles. One-way analysis of variance (ANOVA) used lesion volume as the dependent variable and the site of ablation as a factor to compare the lesion volume distributions between groups. 

To perform multiple linear regression analysis, we verified if the independent variables had explanatory power by checking the beta value, which is the regression coefficient of each independent variable. Multiple regression analysis using the backward elimination technique was used to predict lesion volume. 

A univariate logistic regression analysis was performed to evaluate the relationship between the predictor variables (reported with their respective odds ratios (ORs) and 95% confidence intervals (CIs)) and the presence of extracardiac lesions. 

Statistical significance was defined as *p* < 0.05. All analyses were performed with SPSS Statistics v24.

## 3. Results

Eighteen swines were used for the experiment, nine in each group. The flowchart of this study is shown in Figure 5.

Four RF applications were performed for each swine (one died after only two RF applications). A total of 70 RF applications were performed (36 in the LE group and 34 in the HE group). Two swines in the HE group died immediately after RF delivery from cardiac tamponade, allowing EGM but not MRI analysis.

At the time of histological analysis, 10 specimens in the LE group and 6 in the HE group were found to be insufficient for analysis and, therefore, did not undergo MRI evaluation. Among the samples collected and analyzed, the average sample thickness was 4.30 mm (±1.60) and 4.97 mm (±2.28) in the LE and HE groups, respectively (*p* = 0.333).

### 3.1. Ablative Procedure and EGM Analysis

The characteristics of RF applications are summarized in Table 1. 

As expected from the protocol, the LE and HE groups differed significantly in temperature (34.0 °C vs. 37.5 °C, *p* < 0.001) and duration of RF application (50.0 vs. 20.0 s, *p* < 0.001), but not in force (9.5 vs. 10.0, *p* = 0.717). LSI was significantly lower in the LE group than in the HE group (4.12 ± 0.50 vs. 5.5 ± 1.18, *p* < 0.001).

HE was associated with a greater EGM reduction (0.72 vs. 0.42 mV, *p* < 0.001) and a significantly lower post-ablation R-wave amplitude (0.07 vs. 0.16 mV, *p* = 0.009). There was no difference in the disappearance of the unipolar S wave (19% vs. 16%, *p* = 0.794) after ablation.

The median variation (∆) in EGM amplitude before and after RF application was 0.65 (0.10–1.09) vs. 0.63 (0.22–1.05) for the LE and HE groups, respectively (*p* = 0.990). The median variation in the bipolar R wave (∆R) was 0.19 (0.02–0.56) vs. 0.19 (0.08–0.49) (*p* = 0.866) and the median variation in the unipolar S wave (∆S) was 0.40 (0–0.96) vs. 0.48 (0.04–0.95) (*p* = 0.825), in the LE and HE groups, respectively.

### 3.2. MRI Volume and Qualitative Analysis

Figure 6 shows an example of a 3D MRI reconstruction of an RF lesion. The MRI analysis for each group is summarized in Table 2. 

Overall, the mean RF lesion volume was 140.7 mm^3^ (±135.3) with wide variability. The mean lesion volume was 100.2 mm^3^ (±81.2) in the LE group vs. 178.3 mm^3^ (±163.7) in the HE group (*p* = 0.033). The two groups showed significantly different mean lesion volumes, as reflected in the variance in each group, which was 6.6 mm^3^ in the LE group and 40.5 mm^3^ in the HE group. 

There was no difference in lesion volume variability regardless of whether the atrial tissue was trabeculated or flat (143.8 ± 169.3 mm^3^ in the former group and 158.5 ± 171.8 mm^3^ in the latter group, *p* = 0.784).

In the multivariate linear regression model (Table 3), HE application correlated significantly with greater lesion volume (*p* = 0.037). RF application in the left atrium was also associated with greater lesion volume, albeit not significantly. In contrast, the application of RF to trabecular surfaces showed an inverse and nonsignificant association with lesion volume.

### 3.3. Complications and Extracardiac Lesions

All procedural complications (n = 7) occurred in the HE group (*p* = 0.002) and included the following: steam pop (n = 1), tamponade (n = 4), and ventricular arrhythmias (n = 2).

Two animals died after the procedure, both in the HE group. One died from ventricular fibrillation refractory to defibrillation and the other from cardiac tamponade immediately after the procedure, despite prompt pericardial drainage.

The volume distribution was similar in pigs with and without procedural complications (*p* = 0.762).

Extracardiac lesions were found in 14 sites in the LE group (53.8% of the application), and in 18 sites in the HE group (64.3% of the application) (*p* = 0.613). Detailed results are provided in Appendix A.

The median lesion volume in the group of swines with extracardiac lesions was 174.1 mm^3^ (±167.1), compared with 137.3 mm^3^ (±169.6) in the group without evidence of extracardiac lesions. Volume distributions were similar in swines with and without extracardiac lesions (*p* = 0.240) (Appendix A).

The univariate logistic regression analysis showed that trabeculated atrial tissue was the only variable significantly associated with a higher likelihood of extracardiac lesions (OR 3.667, 95% CI 1.141–11.787, *p* = 0.029) (Appendix A). 

## 4. Discussion

The exact volume of RF lesions produced in the atrium following high-power/short-duration and short-power/long-duration RF ablative strategies is not yet known in the literature. To our knowledge, our study is the first to evaluate RF lesion volume using high-resolution magnetic resonance imaging (9.4 Tesla MRI) on a post-mortem swine model. Our results showed a wide variability in RF lesion volumes, which could not be easily predicted a priori, even under highly controlled clinical-like experimental conditions. RF lesions created using the HE strategy were significantly larger and had a wider volume distribution compared with those from the LE strategy.

Significant progress has been made in recent years in the development of new technologies for better lesion creation, such as contact force and different energy sources [23]. Currently, RF is the most widely used method in clinical practice. However, few data are available on the quantification of lesions produced by RF, and the available data are mostly derived from in vitro models rather than in vivo animal studies conducted days after the ablative procedure [7,19]. Thus, the direct outcomes of RF on viable tissues remain largely unknown, particularly concerning the final extent of the injury, tissue remodeling in response to RF (such as initial edema formation and subsequent fibrous replacement) [24], and the time interval required for lesion maturation. 

The classical histopathological method destroys the specimen, allows for only unidirectional cutting, and is limited by the number of planes measured. Consequently, calculating lesion volume can only be approximate. In contrast, MRI—and particularly high-resolution MRI—is a noninvasive diagnostic method that enables accurate and repeatable quantitative and qualitative assessment of the actual RF lesion volume [25]. MRI allows for a multiaxial and 3D assessment of lesion volume, closely approximating the true volume. Furthermore, MRI analysis a few days post-ablation offers a more precise assessment of the exact RF lesion volume after the initial edema reabsorption [18,24,26]. 

We compared high-power/short-duration with low-power/long-duration RF applications at four atrial sites. In this study, we directly compared these two ablative strategies, which have recently gained popularity in clinical practice and sparked significant scientific debate regarding their effectiveness. The concept of short-duration/high-power ablation for atrial fibrillation has garnered interest because of the potentially shorter procedure time, shorter total RF time, and better transmural lesion formation. However, there is no rigorous definition, and data are often derived from in vitro studies with power settings of 40–90 W with application durations mostly shorter than 15 s [27,28,29]. There also seems to be a generally lower rate of first-pass isolation of pulmonary veins [30,31]. Notably, lesions created by high-power/short-duration ablation are broader [9,32].

Our results showed a wide volumetric variability in the effects of RF. The mean RF lesion volume was 147 mm^3^ (±160) with wide variability in each group, as indicated by the variance of 6.6 mm^3^ in the LE group and 40.3 mm^3^ in the HE group. Lesion volume in the HE group was approximately 35% higher than in the LE group, and the HE strategy was significantly associated with higher lesion volume (coefficient B of 0.403, *p* = 0.037). In contrast, lesion volume was not affected by either atrial location or the type of trabecular/flat surface of the atrial endocardium.

Enomoto et al. [19] demonstrated in vitro that the HE strategy results in larger but shallower lesions, with volumes comparable to conventional RF, a result also noted by Bourier et al. [9]. Other studies suggest that HE application may avoid remote conductive heating, potentially reducing the risk of collateral damage to extracardiac structures [9,33,34]. While these results align with our findings, the following differences should be noted: the different lesion volumes observed may depend on the in vivo setting of our study and the longer duration of RF application. We observed larger lesion volumes and greater variability in the HE group, along with a larger (though not significant) number of extracardiac lesions. This might be due to catheter stability—if the catheter remains stable, a larger lesion can be created; however, poor stability may result in insufficient lesion formation. This explanation has already been explored in the literature with variable results [35,36], and further studies are needed to confirm our hypothesis. Additionally, the possible non-uniform diffusion of thermal energy could be influenced by the vascularization of the atrial wall and the cooling effect of surrounding vessels, although this remains theoretical.

A higher LSI value was also observed in the HE group compared with the LE group, consistent with the larger lesion volume in the HE group. However, in the multivariate analysis, LSI did not correlate with RF lesion volume.

The HE applications led to a significantly greater reduction in intracavitary bipolar signals (TEA and R wave). However, contrary to the findings of Bortone et al., which indicated that modifications in the unipolar atrial EGM after ablation could be useful for monitoring RF application transmurality in clinical settings [37], we did not observe additional value from the S wave unipolar signal or any correlation with lesion volume.

Extracardiac injury was found in 53.8% and 64.3% of applications in the LE group and the HE group, respectively. Although this difference was not statistically significant, it aligns with the observation that most complications occurred in the HE group. Additionally, the trabeculated surface of the atrial endocardium, which is more prevalent in the left atrium of the porcine model, was associated with a higher likelihood of extracardiac lesions (*p* = 0.029). Physicians are generally aware of clinically significant extracardiac injuries, such as tamponade or esophageal fistulas, though they are rare. However, the high incidence of nonclinical extracardiac injuries, particularly in the lungs, pulmonary artery, or aorta, should also be noted. Given the anatomical differences between humans and swines, our findings may not be directly translatable. Nonetheless, it is important for operators to consider these percentages. Hence, although the applications were conducted under conditions similar to “clinical” ones, the four RF points used in a single procedure are significantly fewer than the number typically used in human patients. This disparity could potentially increase the extent of injury in real-world conditions. Consequently, some postoperative symptoms often labeled as “pericardial reactions” might actually be related to extracardiac injuries. While this is a hypothesis, it suggests a need for caution in using high-energy applications that may result in larger and more variable lesions because of shorter application times.

### Limitations

Firstly, our study was limited by the small number of animals; a larger sample size would have been needed to achieve statistical significance in group comparisons, though our sample is still comparable to previous studies on this topic. Secondly, the animals were euthanized after 7 days. The results might differ after several weeks, as lesion stabilization typically marks the end of the remodeling process following RF injury [38]. Unfortunately, our MRI protocol did not allow for further investigation, particularly to assess residual viability within the lesion volume or in the border zone. Thirdly, although the RF applications were carried out by a single operator, the investigator was aware of the ablative strategy used, which could introduce bias. Lastly, our study employed a very specific methodology, including PFA for specimen fixation and a single type of contact force irrigated catheter. Therefore, it is possible that different equipment might have yielded different results.

To our knowledge, this is the first study to use a 9.4 Tesla MRI, and the protocol was designed and implemented entirely in our center. Further studies and refinements in the protocol could enhance the accuracy of volume measurement and qualitative analysis of RF injury.

Our study faced interruptions due to the COVID-19 lockdown, with the first set of experiments concluding in early March 2020. The samples were promptly fixed in PFA and stored until October 2020, after which they were analyzed during the second session of experiments. This delay might have affected sample quality, though the number of unusable samples was similar across both experiment sessions.

## 5. Conclusions

High-resolution MRI allows precise quantification of radiofrequency lesion volume. The high-energy, short-duration RF application shows a significantly larger lesion at the cost of greater variability in lesion volume. Modern ablative techniques (including cryoablation and electroporation) should be studied in terms of the predictability of the lesion to make the techniques reproducible with each other and to better characterize the cause-effect relationship.

## Figures and Tables

**Figure 1 jcm-13-05153-f001:**
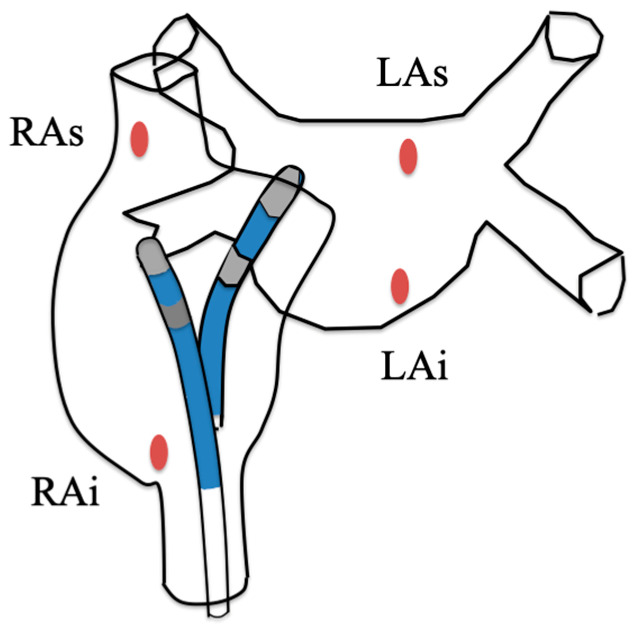
Schematic representation of the four different locations (in red) of RF application performed in each swine.

**Figure 2 jcm-13-05153-f002:**
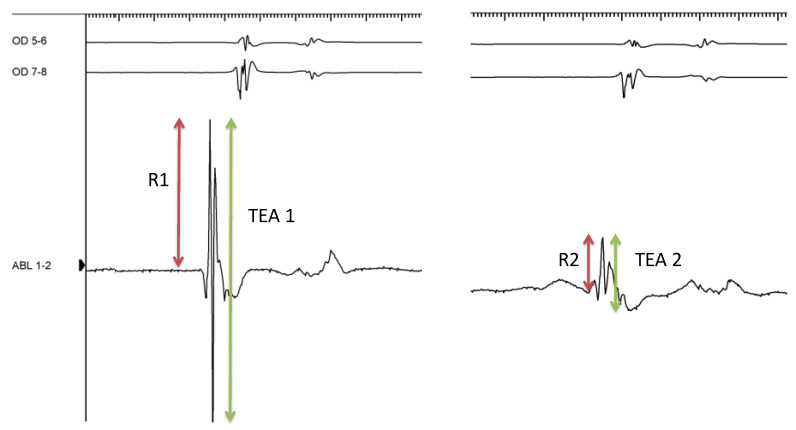
Representative example of bipolar R wave and total electrogram amplitude (TEA) measurements before (**left**) and after (**right**) RF application. OD, right atrium; ABL, ablator catheter signal.

**Figure 3 jcm-13-05153-f003:**
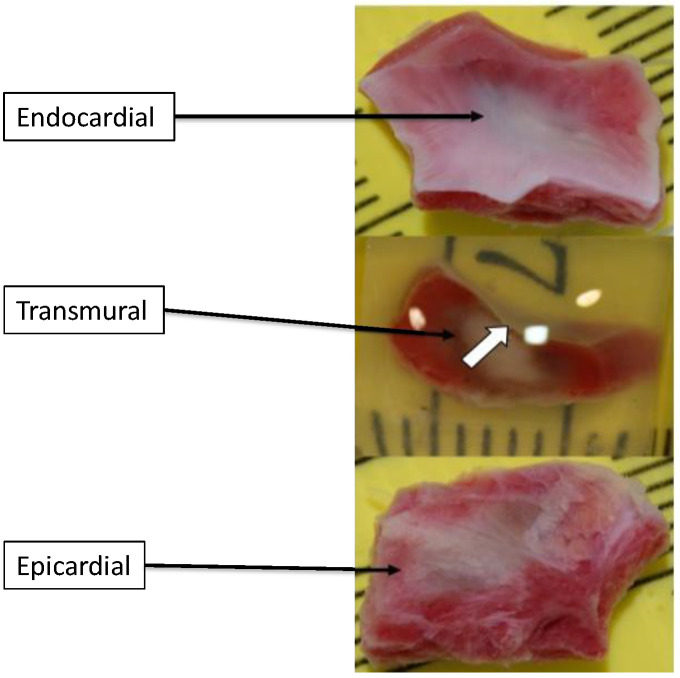
Representative example of excised RF lesion before PFA staining. White arrow: endocardium.

**Figure 4 jcm-13-05153-f004:**
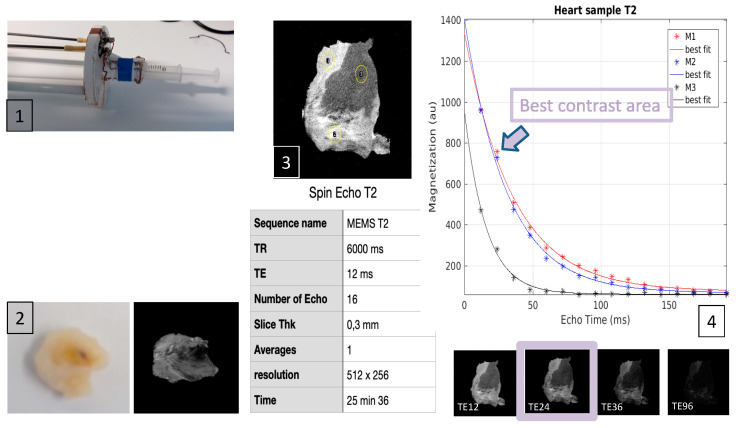
Instrumentation and specific acquisition sequences of MRI imaging (Fast Spin echo 3D). Panel 1: 2 antenna channels and tube into which the tissue sample is inserted for acquisitions. Panel 2: tissue samples after PFA staining and corresponding MRI image. Panel 3: Sequence used (**lower** part) and corresponding image (**upper** part). Panel 4: Different contrasts obtained with the different TEs tested. TE: time of echo, TR: repetition time.

**Figure 5 jcm-13-05153-f005:**
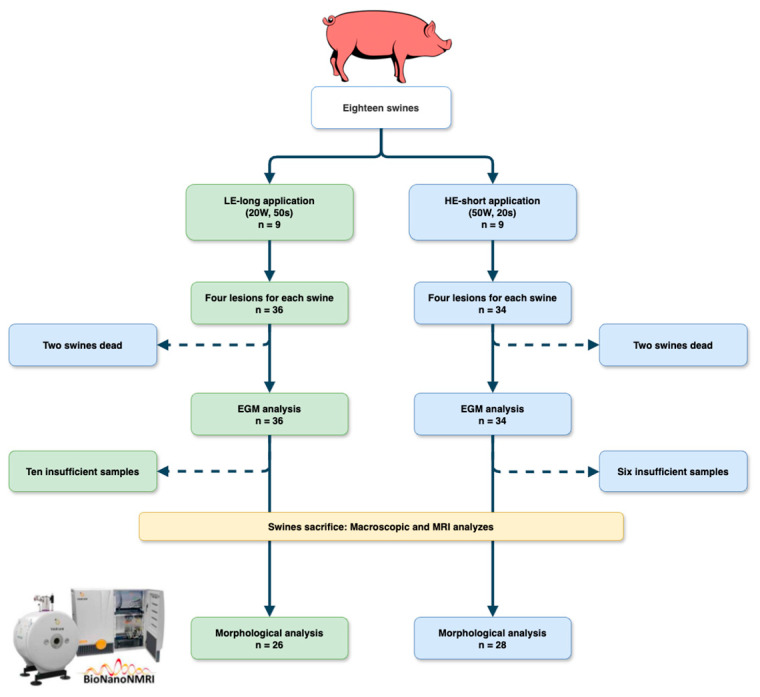
Flowchart of this study. LE, low energy; HE, high energy; EGM, electromyographic; MRI, magnetic resonance imaging.

**Figure 6 jcm-13-05153-f006:**
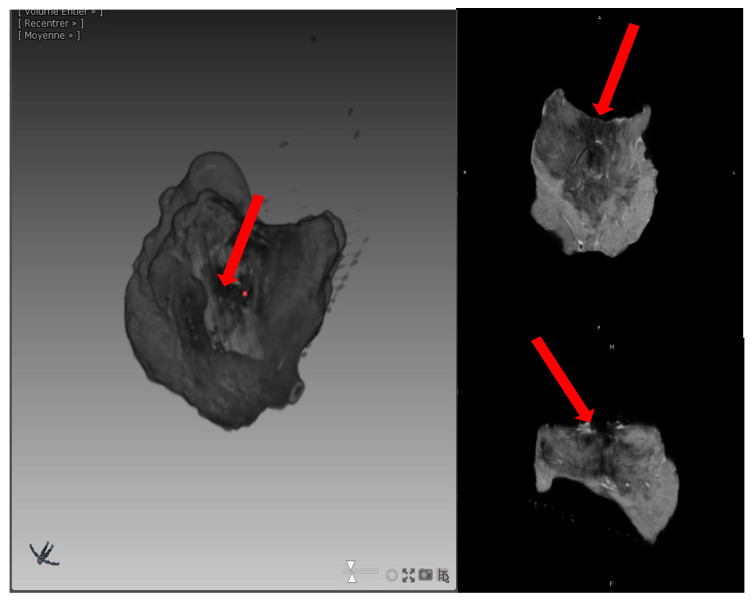
Representative example of a 3D acquisition of an RF lesion. Red arrow: endocardial side with the imprint of the RF catheter.

**Table 1 jcm-13-05153-t001:** Comparison of ablation parameters and EGM signal amplitude results between the LE and HE groups.

	Low Power (20 W) n = 36	High Power (50 W) n = 34	*p* Value
Average impedance (ohm) during RF, median (IQR)	100.5 (96.2–110.7)	91.5 (89.0–98.5)	**<0.001**
Temperature (°C), median (IQR)	34.0 (32.0–35.7)	37.5 (35.0–40.0)	**<0.001**
Duration of ablation (seconds), mean (±SD)	50.0 (±0.17)	20.0 (±0)	**<0.001**
FTI, mean (±SD)	476.28 (±130.20)	237.0 (±143.11)	**<0.001**
LSI, mean (±SD)	4.12 (±0.50)	5.5 (±1.18)	**<0.001**
Force (g), median (IQR)	9.5 (8.0–11)	10.0 (7.0–13.0)	0.717
Amplitude bipolar R wave (mV) before ablation, median (IQR)	0.38 (0.27–0.88)	0.33 (0.15–0.54)	0.153
Amplitude bipolar R wave (mV) after ablation, median (IQR)	0.16 (0.62–0.37)	0.07 (0.02–0.16)	**0.009**
Amplitude unipolar S wave (mV), before ablation, median (IQR)	0.78 (0.22–1.19)	0.77 (0.36–1.34)	0.656
Amplitude unipolar S wave (mV) after ablation, median (IQR)	0 (0–0.28)	0 (0–0.40)	0.811
S-wave disappearance post-RF, n (%)	19 (61.3)	16 (55.2)	0.794
Total EGM amplitude pre-ablation (mV), median (IQR)	1.38 (0.99–1.92)	1.03 (0.67–1.43)	0.052
Total EGM amplitude post-ablation (mV), median (IQR)	0.72 (0.38–1.18)	0.42 (0.22–0.62)	**<0.001**

IQR: interquartile range; mV: milli volts; EGM, electromyographic; FTI: Force–Time Integral; LSI: Lesion Index; SD: standard deviation; RF, radiofrequency; W: watts. Bold: Usually used to highlight statistical significance.

**Table 2 jcm-13-05153-t002:** MRI characteristics of each group.

	Low Power (20 W) n = 26	High Power (50 W) n = 28	*p* Value
Lesion volume (mm^3^), mean (±SD)	100.2 (±81.2)	178.3 (±163.7)	**0.033**
Lesion depth (mm), mean (±SD)	2.78 (±0.85)	3.17 (±1.60)	0.290

SD: standard deviation; IQR: interquartile range; W: watts.

**Table 3 jcm-13-05153-t003:** Predictors of radiofrequency (RF) lesion volume from the linear regression analysis.

	Beta Standardized Coefficient (95% CI)	*p* Value
High-energy application	0.403 (8.262; 262.857)	**0.037**
Trabeculated surface	−0.072 (−158.591; 108.457)	0.706
RF application in left atrium	0.074 (−107.595; 156.891)	0.708

## Data Availability

The data supporting this article will be shared on reasonable request to the corresponding author.

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
