# Peer review of "Radiofrequency Lesion in the Atrial Wall: How Variable Is It? 9.4 Tesla MRI Analysis of Radiofrequency Lesion Volume in a Swine Model"

_jcm, 2024, doi:10.3390/jcm13175153_

Round 1
Reviewer 1 Report
Comments and Suggestions for Authors
The study aims to investigate, using MRI, the volume of radiofrequency lesions in the left atrium and the extracardiac effects resulting from two radiofrequency ablation techniques in piglets — a low-energy long-duration (LE) strategy (20 W, 50 sec) and a high-energy short- duration (HE) strategy (50 W, 20 sec).
I have a few questions:
1. The Statistics section states that the data were presented based on their distribution. However, in the section “MRI volume and qualitative analysis” it's clear that the values of the indicators have a very large dispersion, the average value is less than the standard deviation (SD). For example, the average RF lesion volume is 147 mm3 (± 160), 191.9 mm3 (± 201.1) and others (see pages 8, 9). Such data is completely unreliable. Therefore, it's more informative to present the indicators in the form Me (Q 25%; 75%).
2. In Table 7, according to multivariate linear regression data, a significant relationship was demonstrated only for the “High-energy application” strategy (p = 0.037). However, the text also indicates a direct connection with “RF application in left atrium” and an inverse connection with "Trabeculated surface". That is, the proposal is formulated incorrectly, in fact there is no connection, since the reliability of p significantly exceeds 0.05. In addition, the sentence “The HE application was also significantly associated with higher lesion volume (standardized coefficient (B) 0.403, 95% CI 8.262 to 262.857, p = 0.037)” actually duplicates Table 3, which is redundant.
3. The text contains links to additional tables 1 and 2, additional picture 1, which are not available for viewing.
4. The list of references from 38 sources contains 23 references 5 years ago or more (60.5% - a lot).
5. An important limitation of the study is the small sample size: 9 piglets in each group, 18 piglets in total. The final analysis included data from only 14 piglets (7 in each group).
the authors need to significantly increase the sample size.
Author Response
Comment 1: The Statistics section states that the data were presented based on their distribution. However, in the section “MRI volume and qualitative analysis” it's clear that the values of the indicators have a very large dispersion, the average value is less than the standard deviation (SD). For example, the average RF lesion volume is 147 mm3 (± 160), 191.9 mm3 (± 201.1) and others (see pages 8, 9). Such data is completely unreliable. Therefore, it's more informative to present the indicators in the form Me (Q 25%; 75%).
Response 1: We appreciate the reviewer’s valuable comment. After reviewing the data with our statistics team, we realized that 3 values in the HE group and one value in the LE group were outliers. Therefore, we subjected the lesion volume data to the winsorization process at the 1st and 99th percentiles. In this way, both groups (HE and LE) showed a normal distribution of volumes that allowed this data to be expressed as the mean (±SD). In order to explain this statistical process, we added the sentence “To reduce the influence of outliers, lesion volume data were subjected to winsorization at the 1st and 99th percentiles” to the text. We also updated the data regarding lesion volume. We preferred to delete Figure 7 so as not to confuse the reader.
Comment 2: In Table 7, according to multivariate linear regression data, a significant relationship was demonstrated only for the “High-energy application” strategy (p = 0.037). However, the text also indicates a direct connection with “RF application in left atrium” and an inverse connection with "Trabeculated surface". That is, the proposal is formulated incorrectly, in fact there is no connection, since the reliability of p significantly exceeds 0.05. In addition, the sentence “The HE application was also significantly associated with higher lesion volume (standardized coefficient (B) 0.403, 95% CI 8.262 to 262.857, p = 0.037)” actually duplicates Table 3, which is redundant.
Response 2: We are grateful for this suggestion. We rephrased it as follows:
"In a multivariate linear regression model (Table 3), HE application correlated significantly with greater lesion volume (p = 0.037). RF application in the left atrium was also associated with greater lesion volume, albeit non significantly. In contrast, application of RF to the trabecular surfaces, showed an inverse and nonsignificant association with lesion volume"
Comment 3: The text contains links to additional tables 1 and 2, additional picture 1, which are not available for viewing.
Response 3: We apologize for this inconvenience. All supplementary material has been placed in the file named “manuscript-supplementary.docx” and uploaded of the susy.mdpi system. We re-uploaded the file with additional material (Table S1 and S2 and Figure S1) to the system.
Comment 4: The list of references from 38 sources contains 23 references 5 years ago or more (60.5% - a lot).
Response 4: Thank you very much for your valuable suggestion. We replaced references 2-4 and 24 with newer ones.
Comment 5: An important limitation of the study is the small sample size: 9 piglets in each group, 18 piglets in total. The final analysis included data from only 14 piglets (7 in each group).
the authors need to significantly increase the sample size.
Response 5: We appreciate the reviewer’s comment. We are aware that the sample size is not large, and this was also stressed in the limitations section. However, the sample size is comparable to that of many large animal studies in fundamental research. Unfortunately, the study was conducted close to the pandemic period by Covid-19, with limitations to the conduct of the study well known. However, we believe that already the data collected can provide accurate information that HE radiofrequency results in volumetrically larger lesions with wide variability, more complications, and more extracardiac lesions. We believe that this should prompt the scientific community to study in the same way other sources of energy (such as cryoablation and electroporation) that are widely used now, before making wide clinical use of them.
Reviewer 2 Report
Comments and Suggestions for Authors
The article entitled “Radiofrequency lesion in atrial wall: how variable is it?” is interesting and well-documented and the results on a large scale could have the potential to influence the future targer therapies. However, some adjustments are needed in order to improve the quality of the abstract:
1. First of all, it woul be easier if the authors will use the Journals’ template. Tables and figures should be included in the manuscript, where the authors mentioned about them for a better reading and understanding. Also, references are not written correctly according to the MDPI instruction. Please revise.
2. All figures are important for a better understanding of the protocol. Please put them in the manuscript.
3. In the discussion section, I would suggest to discuss more about the clinical and therapeutical implication of this therapy in human models (maybe talk about some risk scores, the importance of using this therapy vs optimal medical therapy, when this shoul be used etc).
4. Are there studies that compare radiofrequency, cryoablation and medical therapy? It would be interesting to discuss.
Comments on the Quality of English LanguageJust minor English editing for a more scientific soundness.
Author Response
Comment 1: First of all, it woul be easier if the authors will use the Journals’ template. Tables and figures should be included in the manuscript, where the authors mentioned about them for a better reading and understanding. Also, references are not written correctly according to the MDPI instruction. Please revise. All figures are important for a better understanding of the protocol. Please put them in the manuscript.
Response 1: We thank the reviewer for the valuable advice. We have formatted the text and references according to the MDPI format. Regarding the additional material (tables and figures), we apologize for the inconvenience and have re-uploaded the additional material (Table S1 and S2 and Figure S1) to the submission system.
Comment 2: In the discussion section, I would suggest to discuss more about the clinical and therapeutical implication of this therapy in human models (maybe talk about some risk scores, the importance of using this therapy vs optimal medical therapy, when this shoul be used etc).
Response 2: Thank you for the valuable advice. The central role of transcatheter ablation in major arrhythmias, and atrial fibrillation in particular, is now an established principle in the literature and also supported by the most recent versions of the guidelines. We believe that the purpose of our work is not to perform a systematic review of the use of RF in the treatment of atrial fibrillation, but to volumetrically evaluate the RF lesions that are produced with two ablation protocols (HE vs. LE). Therefore, we decided not to dwell on the comparison between RF and medical therapy, thinking that this might divert the reader's attention from the central message of the study.
Comment 3: Are there studies that compare radiofrequency, cryoablation and medical therapy? It would be interesting to discuss.
Response 3: We thank the reviewer once again. To our knowledge, there are currently no studies that have compared lesion volume with RF and cryoablation. There are reports (such as Parvez's work “Comparison of Lesion Sizes Produced by Cryoablation and Open Irrigation Radiofrequency Ablation Catheters,” just to name one), but these are dated, do not quantify lesion volume with MRI, and are not comparable with the experimental conditions proposed in our study. On the other hand, the two forms of energy can be used in different modalities and thus a direct comparison RF and cryoablation would be difficult. There are data in the literature regarding the long-term safety and efficacy of the two ablative strategies. Instead, we believe that it is beyond the scope of our study to compare the efficacy and safety of RF or cryoablation with respect to medical therapy. This might distract the reader from the main message of the study, which is the volumetric quantification of the lesion produced by RF under controlled experimental conditions and in an animal model.
Reviewer 3 Report
Comments and Suggestions for Authors
The study is interesting and adds further insights to the biophysics of RF lesions performing different ablation settings.
some comments:
- 9.4 Tesla MRI is not common even in experimental studies. Please provide the clinical applicability of this imaging system, any possibile collateral effects or safety issues?
- ultra-high resolution MRI probably provides 3D features of RF lesions. Did the authors perform direct comparision (lesion volume) with conventional histopathological analysis? If not, was this kind of Tesal detecting RF lesions validated in prior studies (please cite if needed).
- Why the authors selected those sites to apply RF? easier to do atrial incisions after? Tissue features at the PV-LA junction may be different with different response to RF.
Author Response
Comment 1: 9.4 Tesla MRI is not common even in experimental studies. Please provide the clinical applicability of this imaging system, any possibile collateral effects or safety issues?
Response 1: We appreciate the reviewer's valuable comment. For many years, 9.4 Tesla has been used mainly in preclinical studies. The 9.4 Tesla MRI is used by scientists and is not available for medical or clinical applications. A human being or a large animal cannot fit in the machine, so we had to perform the MRI on a tissue sample and not on a whole animal. In our study, this instrumentation allowed accurate and close-to-real 3D reconstruction of the lesion volume after RF.
Comment 2: Ultra-high resolution MRI probably provides 3D features of RF lesions. Did the authors perform direct comparision (lesion volume) with conventional histopathological analysis? If not, was this kind of Tesal detecting RF lesions validated in prior studies (please cite if needed).
Response 2: We thank the reviewer once again. 3-D histopathological reconstruction was considered for this work, but it would have required many slices to be sufficiently accurate and thus was not cost- and time-effective. In addition, a larger number of animals would have been required for comparison. In addition, clear delineation of the lesion after 7 days would have been questionable with conventional histological staining. The histopathological method does not allow a correct quantification of lesion volume, but only an approximation of it. In calculating the volume, we could have made an approximate calculation starting from the depth of the lesion and calculating the volume as that of a hemisphere. However, we thought it appropriate not to include this calculation in the article so as not to distract the reader's attention from the main result of our study and that is the comparison of RF lesion volume with two different ablative strategies. For the same reasons, we did not perform the comparison between MRI and histological method in volumetric quantification, which has been covered in other papers (such as this one, https://www.ahajournals.org/doi/full/10.1161/01.CIR.102.6.698).
In the discussion section, we explained the above reasoning in the sentences “The classical histopathological method destroys the specimen, allows only unidirectional cutting, and is limited by the number of planes measured. Therefore, the calculation of lesion volume can only be approximate. In contrast, MRI - and particularly high-resolution MRI - is a noninvasive diagnostic method that allows accurate and repeatable quantitative and qualitative assessment of the true RF lesion volume [25]. In fact, MRI allows a multiaxial and 3D assessment of the lesion volume, with a good approximation of the actual volume.”
Regarding the validation of lesion volume by 9.4 Tesla MRI, to our knowledge there is no published work in the literature on the subject. In fact, we believe that this may be the innovative and original aspect of our work.
Comment 3: Why the authors selected those sites to apply RF? easier to do atrial incisions after? Tissue features at the PV-LA junction may be different with different response to RF.
Response 3: We appreciate the reviewer’s valuable comment. Exactly, the lesions were performed at easily recognizable locations in the porcine atrium so that lesions could be more easily localized at the time of excision. In addition, the aim of the study was to achieve, as far as possible, a minimum contact force of 10 g and a maximum of 20 g (as described on page 3, line 97). Therefore, the lesions thus localized in the porcine right and left atrium served this purpose. We are aware that the results might be different if applied to the ostium of the pulmonary veins. However, we believe that the data from our study may answer some important questions, and primarily the safety of applying RF at different sites of the atrium where the risk of complications and extracardiac injury is also higher. Furthermore, our study did not aim to investigate the efficacy of RF ablation regarding atrial arrhythmias such as atrial fibrillation.
Round 2
Reviewer 1 Report
Comments and Suggestions for Authors
It is necessary to correct the values 147 m3 (± 160) (page 1, line 22), 137.3 m3 (± 169.6) (page 9, line 246), 147 m3 (± 160) (page 10, line 307).
Author Response
We sincerely thank the reviewers for their work, which we believe has greatly improved our manuscript.
Regarding the correction of the 147 m3 (± 160) value (page 1, line 22 and page 10, line 307), this refers to the average lesion volume of the study, including both pigs in the LE and HE groups. Since the result could be misleading, we preferred to remove it from the abstract. Instead, we left it in the results and discussed in detail.
On the contrary, the value of 137.3 m3 (± 169.6) found on page 8, line 329 in the paragraph “The median lesion volume in the group of swines with extracardiac lesions was 174.1 mm3 (± 167.1), compared to 137.3 mm3 (± 169.6) in the group without evidence of extracardiac lesions” refers to the comparison of lesion volumes in the group of swines that did or did not show extracardiac lesions (both LE and HE group).
We remain available for any further clarification and improvement that would enhance the work.
Reviewer 2 Report
Comments and Suggestions for Authors
The authors made the necessary changes and the quality of the manuscript is impoved.
Author Response
We would like to thank the reviewer once again for their work, which we are sure has greatly improved the quality of our manuscript.